# Presence of *Listeria monocytogenes* in Ready-to-Eat Artisanal Chilean Foods

**DOI:** 10.3390/microorganisms8111669

**Published:** 2020-10-27

**Authors:** Fernanda Bustamante, Eduard Maury-Sintjago, Fabiola Cerda Leal, Sergio Acuña, Juan Aguirre, Miriam Troncoso, Guillermo Figueroa, Julio Parra-Flores

**Affiliations:** 1Environmental and Public Health Laboratory, Universidad del Bío-Bío, Regional Secreatariat of the Ministry of Health in Maule, Talca 3461637, Chile; ferbustamantetecmed@gmail.com; 2Department of Nutrition and Public Health, Universidad del Bío-Bío, Chillán 3800708, Chile; emaury@ubiobio.cl; 3Department of Food Engineering, Universidad del Bío-Bío, Chillán 3800708, Chile; fcerda@ubiobio.cl (F.C.L.); sacuna@ubiobio.cl (S.A.); 4Department of Agricultural Industry and Enology, Universidad de Chile, Santiago 8820808, Chile; juan.aguirre@uchile.cl; 5Microbiology and Probiotics Laboratory, Institute of Nutrition and Food Technology, Universidad de Chile, Santiago 7830490, Chile; troncoso56@gmail.com (M.T.); gfiguerog@gmail.com (G.F.)

**Keywords:** *Listeria monocytogenes*, artisanal ready-to-eat foods, food safety, antibiotic resistance profile, virulence, premature stop codons

## Abstract

Ready-to-eat (RTE) artisanal foods are very popular, but they can be contaminated by *Listeria monocytogenes*. The aim was to determine the presence of *L. monocytogenes* in artisanal RTE foods and evaluate its food safety risk. We analyzed 400 RTE artisanal food samples requiring minimal (fresh products manufactured by a primary producer) or moderate processing (culinary products for sale from the home, restaurants such as small cafés, or on the street). *Listeria monocytogenes* was isolated according to the ISO 11290-1:2017 standard, detected with VIDAS equipment, and identified by real-time polymerase chain reaction (PCR). A small subset (*n* = 8) of the strains were further characterized for evaluation. The antibiotic resistance profile was determined by the CLSI methodology, and the virulence genes *hlyA, prfA,* and *inlA* were detected by PCR. Genotyping was performed by pulsed-field gel electrophoresis (PFGE). *Listeria monocytogenes* was detected in 7.5% of RTE artisanal foods. On the basis of food type, positivity in minimally processed artisanal foods was 11.6%, significantly different from moderately processed foods with 6.2% positivity (*p* > 0.05). All the *L. monocytogenes* strains (*n* = 8) amplified the three virulence genes, while six strains exhibited premature stop codons (PMSC) in the *inlA* gene; two strains were resistant to ampicillin and one strain was resistant to sulfamethoxazole-trimethoprim. Seven strains were 1/2a serotype and one was a 4b strain. The sampled RTE artisanal foods did not meet the microbiological criteria for *L. monocytogenes* according to the Chilean Food Sanitary Regulations. The presence of virulence factors and antibiotic-resistant strains make the consumption of RTE artisanal foods a risk for the hypersensitive population that consumes them.

## 1. Introduction

Foodborne diseases are a major public health problem worldwide. They have an impact on consumer health and the economies of importing countries [1]. The incidence of foodborne diseases has been increasing because of factors such as changes in industrial production, food globalization, new economic treaties, and more recently, the need to consume fresher, healthier, and ready-to-eat (RTE) foods with minimal or moderate production and artisanal processing [2]. Minimally processed foods are fresh, perishable, unprocessed, and produced by a primary producer or retailer that can market them. Moderately processed foods use pressing, grinding, milling, and other processes to convert food into culinary products for sale in the home, restaurants such as small cafés, or on the street [3].

However, minimally or moderately processed RTE foods can be a health risk for consumers because they could contain dangerous microorganisms [4]. Inadequate manufacturing and hygiene practices throughout the RTE process can lead to contamination by and/or survival of microorganisms. In addition, this can promote the proliferation of microorganisms and reach harmful levels for health if temperature control is inadequate during the distribution and/or marketing of RTE products [5].

One of the main bacteria responsible for the loss of food safety in RTE foods is *Listeria monocytogenes,* which causes up to 65% of food recalls [6]. This pathogen is ubiquitous and persistent in processing rooms and raw materials [7]. It causes listeriosis, a potentially severe and invasive illness, which can result in stillbirths, abortions, meningitis, and septicemia [8]. Fatality occurs in up to 30% of cases, and pregnant women, newborns, children, the elderly, and immune-compromised individuals are the high-risk groups that are especially vulnerable [9].

Listeriosis has been a notifiable disease in Chile since 2005, and it is considered as a laboratory surveillance issue according to national sanitary regulations [10]. The microbiological criteria for *L. monocytogenes* in RTE foods were recently incorporated in the Chilean Food Sanitary Regulations (RSA) in 2009. The objective was to promote the surveillance, control, and epidemiological study of cases and outbreaks associated with *L. monocytogenes* [11]. 

A listeriosis outbreak occurred at the end of 2008 and beginning of 2009; it involved 119 cases associated with the consumption of cheese contaminated with *L. monocytogenes* and resulted in 13 fatalities [12]. The number of reported food-related cases of listeriosis between 2015 and 2018 was 75, 65, 83, and 97, respectively, with a mortality rate between 20% and 25% [13]. The most affected population groups were adults over 60 and pregnant women [14]. 

The risk associated with *L. monocytogenes* is still controversial because the infective dose depends on such aspects as the immune status of the host, virulence factors of the pathogen and their respective expression, resistance to adverse environmental conditions, and the effect of the food matrix on the bacterium [15]. Although several authors postulate that a high inoculum is required to become ill with *L. monocytogenes*, counts greater than 100 CFU/g are sufficient to trigger the disease [16]. 

There is scarce information on this pathogen in Chile. Therefore, a study was designed to determine the presence of *Listeria monocytogenes* in Chilean artisanal ready-to-eat foods and evaluate its food safety risk. 

## 2. Materials and Methods 

### 2.1. Sampling

A total of 400 samples of artisanal foods (with minimal and moderate processing) were analyzed because they were considered to be at risk by the health authority. Minimally processed foods are home-produced and moderately processed foods originate in family-owned shops with good manufacturing practices (GMP). Samples included cheeses (*n* = 90), cooked meats (artisanal ham, pâté, sausages, blood sausages) (*n* = 235), pre-processed fruit and vegetables (chopped fruit, fruit salads with strawberries, melon, and peaches, and leafy vegetable salads) (*n* = 35), meals and mixed dishes with raw and/or cooked ingredients (*n* = 40) sampled as part of the Emerging Pathogens Program of the Chilean Health Authority.

### 2.2. Isolation and Enumeration of Listeria monocytogenes

Isolation was performed according to the ISO 11290–1:2017 method with two-stage enrichment. First, 25 g of each food sample was inoculated in 225 mL half Fraser broth (Basingstoke, Oxoid, UK) for initial selective enrichment and homogenized in a stomacher (Seward 400, Radnor, PA, USA). Incubation followed at 30 ± 1 °C for 25 ± 1 h and 0.1 mL of the broth culture was inoculated in 10 mL of full-strength Fraser broth for the second enrichment, which was cultured at 37 °C for 24 ± 2 h. A loopful (1 mL) of each of the half- and full-strength Fraser broths were plated on the chromogenic *Listeria* agar Ottaviani and Agosti (ALOA agar) (Merck, Darmstadt, Germany). The plates were incubated at 37 °C for 24 to 48 h. Five typical colonies from each ALOA agar were restreaked on tryptic soy agar supplemented with 0.6% yeast extract (TSA-YE) (Sigma, Darmstadt, Germany) as a nonselective medium and incubated at 37 °C for 24 to 48 h. The colonies from TSA-YE were verified by Gram staining, catalase reactions, oxidase tests, carbohydrate utilization, CAMP tests, and motility at 20 to 25 °C. *Listeria monocytogenes* was enumerated in the samples according to the methodology described in the ISO 11,290–2:2017 Microbiology of the food chain standard—Horizontal method for the detection and enumeration of *L. monocytogenes* and *Listeria* spp.—Part 2: Enumeration method [17,18].

### 2.3. Detection of Listeria monocytogenes 

After the incubation period, described in the previous point, an immunoenzymatic assay was performed with the VIDAS *L. monocytogenes* equipment (Vitek Immunodiagnostic Assay System, bioMerieux Vitek Inc., Hazelwood, MO, USA) according to the manufacturer’s instructions. Measurement and interpretation were automatically performed by the equipment, reporting the detection as positive or negative according to the validated AOAC (Official Method of Analysis N° 2004.2) protocol for food matrices. 

### 2.4. Phenotypic Identification of Listeria monocytogenes 

Phenotypic identifcation was performed using the API *Listeria* gallery (bioMérieux, Marcy-l’Étoile, France) according to the manufacturer’s instructions. It was incubabed at 35 ± 1 °C for 24 h. The resulting numerical profile was evaluated with the APIWEB program provided by the manufacturer. *Listeria monocytogenes* ATCC 35152 was used as a control.

### 2.5. Molecular Identification of Listeria monocytogenes 

The genomic DNA of the suspected strains was extracted and purified with an UltraClean Microbial DNA Isolation Kit (Mo Bio Laboratories, Qiagen, Carlsbad, CA, USA). Strains were confirmed as *L. monocytogenes* by *hlyA* gene amplification [19] with the Stratagene Mx3000P real-time PCR System equipment (Agilent Technologies, Santa Clara, CA, USA) using the commercial Brilliant II SYBR Green QRT-PCR Master Mix [20] (Agilent Technologies, Santa Clara, CA, USA). Amplified products were sent for sequencing to Macrogen, Korea. They were then analyzed with the Gentle software and assembled with PRABI-Doua (France). The free BLASTn (NCBI) database was used to identify the strains.

### 2.6. Antibiotic Resistance Profile 

The disc diffusion method was applied based on the recommendations by the Clinical and Laboratory Standards Institute [21]. The commercial antibiotic discs consisted of 10 µg ampicillin, 10 µg penicillin, 25 µg sulfamethoxazole-trimethoprim, 15 µg erythromycin, 30 µg vancomycin, and 30 µg chloramphenicol. The strain resistance/susceptibility profiles were characterized by measuring the inhibition zone and interpreting inhibition diameters according to the manufacturer’s instructions and using *S. pneumoniae* ATCC 49619 as a reference microorganism of the CLSI guideline. In addition, *Escherichia coli* ATCC 25922 and *L. monocytogenes* ATCC 7644 were used as controls.

### 2.7. Detection of Virulence Genes

The method described by Aznar and Alarcón [22] was used to amplify conserved regions of three characteristic virulence genes, that is, listeriolysin O (*hlyA*), positive regulatory factor A (*prfA*), and internalin A (*inlA*) (Table 1). The genomic DNA of the suspected strains was extracted and purified with the UltraClean Microbial DNA Isolation Kit (Mo Bio Laboratories, Qiagen, Carlsbad, CA, USA) and mixed with GoTaq Green Master Mix (Promega, Madison, WI, USA) in a thermocycler (Fermelo Biotec, People’s Republic of China). The amplified products of each gene were sent for sequencing to Macrogen, Korea. They were then analyzed with the Gentle software and assembled with PRABI-Doua (France). The *L. monocytogenes* strain ATCC 19115 was used as the positive control, while the *L. innocua* ATCC 33090 strain was used as the negative control. 

The amplified products were sent for sequencing to Macrogen, Korea. Sequences were assembled with the Gentle software and aligned with ClustaW. The consensus sequences were compared with the BLAST database available at https://BLAST.ncbi.nlm.nih.gov/BLAST.cgi.

### 2.8. Serotype and Sequence Type (ST) of Listeria monocytogenes

The *L. monocytogenes* strains were sequenced at the Institute of Medical Microbiology and Hygiene AGES—Austrian Agency for Health and Food Safety, Austria. This information allowed us to determine the serotypes by the extraction of specific sequence targets of the genome [26] using the *L. monocytogenes* 5-plex PCR Serogroup task templates of the SeqSphere+ v. 7.2.0 (2020-7) software with fragments from five DNA regions (*lmo118*, *lmo0737*, ORF2110, ORF2829, and *prs* as an internal amplification control) according to the description by Doumith et al. [27] and Lee et al. [28].

The STs were determined from whole-genome sequencing (WGS) with Task templates for available MLST schemes from the SeqSphere+ v. 7.2.0 (2020-7) software; the ST was determined in the isolates with fragments from the seven housekeeping genes *abcZ*, *bglA*, *cat*, *dapE*, *dat*, *Idh*, and *ihkA* [29,30] and with the profiles of the Institut Pasteur MLST *Listeria* database (http://bigsdb.pasteur.fr/Listeria/Listeria.html).

### 2.9. Pulsed-Field Gel Electrophoresis (PFGE)

Pulsed-field gel electrophoresis (PFGE) was carried out following the Centers for Disease Control and Prevention (CDC) standardized PulseNet protocol for *L. monocytogenes* (www.cdc.gov/pulsenet) [31] with AscI and ApaI as the restriction endonuclease. The PFGE patterns were analyzed with the Gel ComparII Software (Bionumerics 2011 Applied Maths NV), which uses the standard strain *Salmonella* enterica serovar Braenderup H9812 loaded in three lanes in each gel to normalize the images. The strain ATCC 19115 was used as the *L. monocytogenes* control. Matching and the dendrogram from the unweighted pair-group method with arithmetic mean (UPGMA) analysis of the PFGE patterns was performed by the Dice coefficient with a 1.5% tolerance window. 

### 2.10. Estimation of Listeria monocytogenes Concentration at the End of Product Shelf Life

The approach described by Aguirre et al. [32] was used to determine the biological variability of micropopulations and the distribution of *L. monocytogenes* at the end of the shelf life of most of the contaminated food group. Kinetic data were fitted to various distributions with the @Risk 4.5 software for Excel (Palisade Corporation, Newfield, NY, USA). The goodness of fit was compared by three different methods: X2, Anderson Darling, and Kolmogorov–Smirnov. 

The best-fitted distributions, based on the mentioned criteria, were introduced into an exponential model with lag to describe the growth of individually treated and untreated cells using Monte Carlo simulations as described by Koutsoumanis and Lianou [33] expressed as
(1)$$N_{t}=\;\left(N_{o}-\;N_{g}\right)+\;\sum_{1}^{N_{g}}\left\{\begin{array}{ll} 1 & for\;t\leq_{i}\\ e^{{µ}_{max}\;i\;\left(t{-}_{\lambda i}\right)} & for\;t{>}_{i} \end{array}\right.$$
where *N_t_* is the total number of cells at time *t*, *N_o_* is the intial number of cells in the population at *t* = 0, *N_g_* (Binomial *N_o_*, Pg), and Pg is the mean probability of growth [34].

### 2.11. Bioinformatics and Statistical Analyses 

Sequencing products were analyzed with the Gentle software and aligned with ClustaW. A phylogenic tree was constructed by the maximum likelihood method with the MEGA7 software. Central tendency measurements were used for statistical description and the Mann–Whitney test was used for comparison with the STATA 15.0 software with a significance level of α = 0.05. 

## 3. Results

### 3.1. Detection and Quantification of Listeria monocytogenes 

Of the 400 analyzed samples, 30 (7.5%) were positive for *L. monocytogenes* by the VIDAS system. The highest positivity was 17.5% (7/40) for prepared meals and dishes, followed by 8.6% (3/35) for pre-processed fruit and vegetables, 8.5% (20/235) for cooked meats, and 0% (0/90) for cheese and fresh cheese. on the basis of food type, the highest proportion was 11.6% for artisanal foods with minimal processing, which was significantly higher than 6.2% for moderately processed foods (*p* > 0.05) (Table 2). 

As for pathogen quantification, minimally processed RTE artisanal foods such as cooked meats and pre-processed fruit had the highest counts with 5.7 × 10^2^ and 2.2 × 10^3^ CFU/g, respectively. Meanwhile, prepared meals and dishes with moderate processing had the lowest counts with 1.6 × 10^2^ CFU/g (Table 3). In addition, 8.5% of the samples exceeded these limits (100 CFU/g) at the beginning of their shelf life.

Due to economic limitations, only eight strains were selected, four minimally processed (1116, 1259, 1310, and 1383) and four moderately processed isolates (1202, 1263, 1264, and 1310). An in-depth study of phenotypic and genotypic characterization was conducted with these isolates. 

### 3.2. Phenotypic Identification with API Listeria 

The eight selected strains were identified as *L. monocytogenes*. Seven strains had the same API *Listeria* numerical profile, number 6510, and only one strain (1383) had a different number, that is, 6410. This occurred because strain 1383 was unable to acidify D-arabitol (DARL).

### 3.3. Molecular Identification 

All the strains were identified as *L. monocytogenes* when the hly gene was amplified. This was also confirmed with BLASTn, which identified *L. monocytogenes* at a 99% and 100% identification level (Figure 1).

### 3.4. Antibiotic Resistance Profile 

Of the eight analyzed strains, three were susceptible to all the tested antibiotics (1202, 1259, and 1263), three strains had intermediate susceptibility to ampicillin (1116, 1310, and 1540), one strain was resistant to ampicillin (1383), and one strain was resistant to ampicillin and sulfamethoxazole-trimethoprim (1264). 

### 3.5. Virulence Factors 

All the *L. monocytogenes* strains were positive to the three evaluated virulence genes (*hlyA*, *prfA*, and *inlA*). As for sequencing the *inlA* gene, six of the eight *L. monocytogenes* strains exhibited mutations resulting in premature stop codons (PMSC) associated with changes in amino acids. Strain 1259 exhibited a type 6 mutation (Table 3); four new mutations were detected in strains 1116, 1263, 1264, 1310, and 1383, which could not be identified because they are not described or published in the current literature (Figure 2).

### 3.6. Serotype and Sequence Type (ST) 

Strains 1116, 1202, 1263, 1264, 1310, 1259, and 1540 were identified as serotype 1/2a and ST 8, but the ST for strain 1259 could not be determined, making it a potentially new ST. Only strain 1383 exhibited serotype 4b and ST 388.

### 3.7. Pulsed-Field Gel Electrophoresis (PFGE)

Culture purity was initially corroborated by Gram stain and the macroscopic characteristics of the colonies. Two colonies were selected for characterization by PFGE, designated as “a” and “b”, from the samples in which different colonies were detected (D-1263, G-1383, K-1202, L-1310). The PFGE was used to characterize 13 bacterial isolates. In the association dendogram created on the basis of the similarity of DICE between isolates and the UPGMA association method, eight groups of pulse types had similarities greater than 90% and four included more than two bacterial isolates. From the last four groups, two only consisted of isolates from the same sample (L 1310a and L 1310b; K 1202a and K 1202b), one shared strains between samples (G 1383b and H 1540), and one shared isolates from the same and another sample (D 1263a, D 1263b, and E 1264). In addition, strain 1383, which exhibited different morphological colonies as to size and color, had two different pulsetypes. The other three strains in which two colonies were selected for analysis had identical pulsetypes (Figure 3). 

### 3.8. Distribution of Listeria monocytogenes Concentrations at the End of Product Shelf Life

Some 8.5% of samples exceeded the 100 CFU/g limit of *L. monocytogenes* allowed in the Chilean Food Sanitary Regulations at the beginning of their shelf life. The Monte Carlo simulation of the *L. monocytogenes* distribution at the end of product shelf life showed that 39% of RTE cooked meat products can exceed the permitted limits for *L. monocytogenes* (95% CI) at the end of their shelf life (Figure 4).

## 4. Discussion

The importance of wholesome and RTE food for a healthier population has been demonstrated in recent decades [36]. These artisanally processed RTE foods are increasingly popular because of social and environmental interests; however, there is no clear definition or specific regulations associated with them since this type of food is linked to individual producers or very low scale production resulting in dissimilar quality and safety standards [37]. The growing need to consume healthier foods has also produced certain health risks associated with their contamination by different pathogenic microorganisms such as *L. monocytogenes*, which is particularly relevant.

The 7.5% positivity encountered in our study concurs with other studies that have analyzed similar foods with raw or cooked ingredients, pre-processed fruit and vegetables, and minimal or moderate processing (Table 2). In Morocco, Amajoud et al. [38] found *L. monocytogenes* prevalence of 16.6% in salad dressings and 2.7% in beef. Another study of raw materials used in salads and prepared dishes conducted in Iran reported a prevalence of 14% in cucumber, 12% in dressings, 10% in lettuce, and 6% in meat products [39]. Segura and Chávez [40] indicated 8% of RTE fruit salad samples were contaminated with *L. monocytogenes* in Peru. In Turkey, the prevalence of *L. monocytogenes* in RTE vegetables varied between 5% and 15% [41]; this is similar to positivity encountered in Malaysia where 15% contamination was also revealed in this type of food [42]. Studies conducted in the United States demonstrated that *L. monocytogenes* contamination in RTE fruit salad varied between 0.5% and 2.4% [43]. Meanwhile, another study in Spain showed that the presence of *L. monocytogenes* was only 0.7% in minimally processed RTE fruit salad [44]. These prevalence values of *L. monocytogenes* in RTE foods are very similar to those found in our study, except for the United States; our values fluctuated between 17.5% for prepared meals and dishes and 8.5% for pre-processed fruit and vegetables. In addition, these positivity values of *L. monocytogenes* are consistent with the type of manufacturing process for the minimally or moderately processed RTE foods. The presence of *L. monoctogenes* is directly associated with or without the implementation of adequate manufacturing and hygiene practices used by the producers of RTE foods in the different countries where such foods are produced. 

Several authors postulate that a high inoculum is required to become ill from *L. monocytogenes.* However, it has been found that counts greater than 100 CFU of viable cells in the case of high-risk groups, or 10 000 CFU in the case of the healthy population with a strong immune system, would be sufficient to cause the disease [45,46]. Under this premise, if foods are able to bear the growth of *L. monocytogenes*, the possibility of reaching a disease-causing dose is greater. Another empirically-based epidemiological risk model using eight RTE products resulted in infective doses between 10^2^ and 10^6^ CFU/g. Another model using data on the incidence of listeriosis and prevalence of this pathogen in smoked fish obtained a minimal dose that caused listeriosis with 10^4^ CFU/g [47]. In our study, we used the Monte Carlo simulation to estimate that 39% (95% CI) of the evaluated RTE foods would exceed the limit of 100 CFU/g allowed in Chile for products that promote the development of *L. monocytogenes* [11] at the end of their shelf life (Figure 4). In addition, these high counts can increase the risk of disease associated with *L. monocytogenes*, which is a concern if we consider that counts in the initial food evaluation already showed that 8.5% of samples exceeded these limits (100 CFU/g) at the beginning of their shelf life. This shows the need for greater control because there are cases of the disease each year in the province where this study was carried out, and there were two fatal cases in 2019 [48]. A key aspect to consider is that strict safety protocols, such as the HACCP system or GMP, are not applied in the manufacturing process of these types of artisanal RTE foods, which has a direct impact on their final safety [49]. Our results show that the highest counts were in the group of minimally processed RTE cooked meats, and the lowest count was in this same food group with moderate processing, which corroborates our hypothesis of the need to implement a safety system to guarantee safe food for the population. 

Another significant aspect evaluated in our study was the antibiotic resistance profile. The importance of this test is related to the treatment of listeriosis, which still uses antibiotics [50]. In general, most of the *Listeria* spp. isolated from food and clinical samples and environments are sensitive to antibiotic therapy, which is usually used against Gram-positive bacteria, including tetracyclines, ampicillin, penicillin G, imipenem, amoxicillin, sulfonamides, aminoglycosides, macrolides, chloramphenicol, and glycopeptides [51]. It should be noted that bacteria are very versatile in adapting and generating resistance to antibiotics [52]. The results of our study revealed susceptibility to the tested antibiotics, and two strains (1264 and 1383a) were resistant to ampicillin and sulfamethoxazole-trimethoprim (STX). Strains 1116, 1310, and 1540 showed intermediate resistance to ampicillin, which is a concern because previous studies in Chile have indicated that *L. monocytogenes* is susceptible to ampicillin [53]. This is particularly relevant because ampicillin, amoxicillin, and gentamicin are still used to treat listeriosis [54]. Only strain 1264 was resistant to the antibiotic sulfamethoxazole-trimethoprim in the present study (Table 3). However, antibiotic resistance among bacteria transmitted by food, including *L. monocytogenes*, has evolved in recent decades. Yucel et al. [55] reported 66% resistance to ampicillin and STX in strains of *L. monocytogenes* isolated from raw and cooked meats. Conter et al. [56] found resistance to ampicillin in two strains of *L. monocytogenes* isolated from RTE salmon. Jamali et al. [57] observed that the resistance among *L. monocytogenes* strains from fish products was 20.9% to 27.9% resistant to tetracycline and ampicillin, respectively. Kuan et al. [58] tested the antibiotic resistance of 58 *L. monocytogenes* strains from vegetable farms and retail markets in Malaysia and found 5.2% resistance to STX. Arslan and Baytur [59] detected 6.1% resistance to ampicillin and 12.1% to STX in *L. monocytogenes* strains isolated from retail meat This resistance is pertinent because the STX antibiotic is usually used to treat listeriosis in persons who are allergic to penicillin [60]. In addition, the emerging resistance to penicillin of strains from clinical samples poses another major public health problem because penicillin is the gold standard to treat human listeriosis [61]. Therefore, encountering ampicillin and sulfamethoxazole-trimethoprim resistant strains and intermediate resistance to ampicillin in our study should alert authorities and food producers to the potential danger associated with the consumption of these artisanal RTE products. 

Some studies have observed that between 8% and 21% of natural *L. monocytogenes* isolates, both food and environmental, are either attenuated in their virulence or are totally avirulent [62]. In our study, the eight analyzed *L. monocytogenes* strains were positive to the three virulence genes used. This finding is especially relevant for the prognosis and future severity of the disease in individuals who consume contaminated RTE foods and develop the disease from this pathogen, as occurs in this part of Chile. The *hlyA* gene is a virulence factor that encodes for listeriolysin O, a pore-forming cytolysin, which allows entry into host cells [63]; this toxin can be easily detected with blood agar in hemolysis tests [64]. The virulence factor *hlyA* is only present in virulent species of *Listeria* spp. [65], which is frequently used to evaluate the presence of virulence factors in *L. monocytogenes* strains isolated from RTE foods [66]. Virulence factors are regulated by other genes, which are also virulence factors. Thus, the PrfA protein is an absolutely indispensable virulence factor for the expression of virulence in pathogenic species of *L. monocytogenes,* which also depends on environmental conditions such as high temperature (37 °C) and stress conditions [67]. The presence of the *hlyA* and *prfA* genes in all the *L. monocytogenes* strains evaluated in our study constitutes an additional risk for people who frequently consume this type of food. The *inlA* gene, also used in the present study, is another relevant virulence factor for the entry of the bacterium into the organism and which participates in the adhesion process between the bacterial wall and intestinal cells of the individual [68]. The two main invasion proteins of *L. monocytogenes* are InlA and InlB. They are members of a protein superfamily called internalins (Inl), and 41 genes that encode for proteins of this type have been detected in the *L. monocytogenes* genome [69]. A relevant aspect associated with the *inlA* gene is the presence of PMSC. The PMSC have been associated with impaired virulence, which affects their ability to invade human cells. In addition, mounting evidence suggests that all the *inlA* gene PMSC mutations are more frequent in *L. monocytogenes* strains isolated from food than in clinical strains [70]. In our study, only six of the eight evaluated strains exhibited PMSC. We also identified that the *inlA* gene in strain 1259 is a mutation known as type 6 (Table 3). We detected four mutations that could not be identified because they are not described or published in the current literature, which is why we believe they are new mutations (Figure 2). Likewise, a very interesting finding was the presence of mutations in strain 1383 serotype 4b. The PMSC have only appeared in the *inlA* gene in serotype 4b in publications since 2019. Zamani et al. [71] found a strain of *L. monocytogenes* serotype 4b with PMSC. This strain had PMSC in nucleotide 1380, truncated *inlA* gene, and replacement mutation G → A; it is categorized as PMSC number 8. Upham et al. [72] genomically analyzed clinical and food strains and detected that the prevalence of the *inlA* gene PMSC mutations in genomes of food strains was significantly higher (*p* < 0.0001) than in clinical strains. However, a three-codon deletion in the *inlA* gene associated with a highly invasive phenotype was more prevalent in genomes from clinical strains, mainly from serotype 4b, than from food strains (*p* < 0.001). These aspects encountered in the literature are consistent with our reported findings and further in-depth study is required.

Serotype identification is important because not all *L. monocytogenes* serotypes are pathogens. There are currently 13 serotypes of which 4b, 1/2a, and 1/2b are the most frequently associated with the development of listeriosis in more than 98% of human cases [73]; however, other authors include serotype 1/2c [74]. According to our comparative analysis, the strains found in our study belong to serotypes 1/2a and 4b. Toledo et al. [75] analyzed 38 clinical and non-clinical isolates in Chile and found that 65% of the clinical isolates belonged to the serogroup IVb (which includes serotype 4b), while serogroups IIa and IIb represented 17% and 13% of the isolates, respectively. Among the isolates of the non-clinical samples, serogroup IVb was also the most common (33%). Ulloa et al. [76] and Paduro et al. [77] analyzed strains of *L. monocytogenes* isolated in clinical cases and foods and reported that the most common serotype was 4b followed by 1/2a. In South America, Braga et al. [78] found that serotypes 1/2b and 4b were the most prevalent, 45.8% and 41.7%, respectively, in RTE foods in Uruguay. Meanwhile, in Colombia, serotype 4b occurred in more than 50% of the RTE food isolates, followed by 10.8% for 1/2c and 9.5% for 1/2a [79]. In Brazil, various authors have encountered a high prevalence of serotype 4b in RTE foods, while serotypes 1/2a and 1/2c were found in unprocessed and processed RTE foods [80,81,82].

In the present study, ST 8 and ST 388 were identified by MLST; these STs have been found in *L. monocytogenes* isolated from RTE foods such as salmon and poultry products, cooked meats, and vegetables, which are raw materials widely used in sandwiches, salads, and prepared foods sampled in our study [83,84,85]. The use of MLST in *L. monocytogenes* is still limited in South America; Chile and Brazil have reported the use of this molecular typing technique on this pathogen. Chile reported the presence of ST 8 in a clinical case isolate and another in RTE foods in 2018. Ulloa et al. [76] encountered ST 8 in clinical cases and foods and ST 388 was first mentioned in a clinical case. In Brazil, Oxaran et al. [86] reported ST 8 for the first time in cheese commercialized in the country. Camargo et al. [87] detected ST 8 in raw Brazilian beef; when compared with 270 existing ST profiles in South America, they determined that ST 8 had only been reported in Chile and Brazil, and ST 388 was not found in any other country. Several authors state that *L. monocytogenes* ST 8 is one of the most persistent STs in RTE food processing plants. This persistence promotes the risk of recontamination by *L. monocytogenes* in food; future studies are therefore required concerning the factors that confer this persistence in *L. monocytogenes* strains so as to develop preventive control measures [88]. 

Two groups of *L. monocytogenes* were observed when revising the phylogenetic tree of the *hlyA* gene of the strains under study and adding for comparison two *hlyA* genes of *L. monocytogenes* clinical strains. The group of strains 1116, 1202, 1259, 1264, 1310, and 1540 are genetically related, whereas strains 1263 and 1383 are less related to the former group and between strains (66%) when compared with *L. monocytogenes* strains (Figure 1). This genotypic difference could not be corroborated by PFGE because they were different pulsetypes. The PFGE was the gold standard of molecular typing, however, WGS is now considered as the standard. However, PFGE is still used in research because the comparison of the restriction patterns provides information on the relationship between the different isolates. In our study, we observed two unrelated clusters (45.1%) with eight well-related pulsetype subgroups (>90%). As previously mentioned, isolates 1202, 1263, 1310, and 1383 were evaluated in duplicate (referred to as “a” and “b”) because there were small macroscopic morphological differences between them. This variability can be due specifically to the isolation method and subsequent recovery that was used; however, the duplicate strain sent for WGS did not show a mixture of strains according to the initial information provided by the institution that sequenced the genome of our *L. monocytogenes* strains. Therefore, strains 1202, 1263, and 1310 exhibited an identical pulsetype (Figure 3). The remaining strain (1383) originating in cooked meats exhibited two unrelated pulsetypes. The pulsetype of strain 1383a was more related to the pulsetype of strain 1310 from pre-processed fruit, while the pulsetype of strain 1383b was related to the pulsetype of strain 1540, both from cooked meats. Another study conducted in Chile analyzed 94 pulsetypes of *L. monocytogenes* and we detected a great electrophoretic similarity when we compared the ST 8 and ST 388 strains from that study [76] with our strains 1263 and 1383, that is, the same STs and serotypes. This scenario suggests the persistence of *L. monocytogenes* over time, as described by other authors such as Kudsen et al. [88]. Likewise, WGS technologies emerge as the best typing tool for *L. monocytogenes* isolates in epidemiological research. Sequencing the whole bacterial genome provides an unparalleled depth of information to compare between species and subspecies and generate information on aspects such as virulence factors, sequence types, and antimicrobial resistance genes. This type of sequencing is far more comprehensive than the insight provided by PFGE [89]. 

The limitation of this study is related to the use of only eight strains of *L. monocytogenes* due to economic reasons. This situation restricts our knowledge of the phenotypic and genotypic characteristics of *L. monocytogenes* in the foods under study, which are associated with disease and mortality in Chile. However, these eight strains were randomly selected and represented eight independent samples found positive for *L. monocytogenes* from producers who manufacture different artisanal RTE foods. Although it is not possible to extrapolate this information to the positives of the analyzed samples, it provides us with a focused approach to reality regarding the source, quantification, virulence, and antibiotic resistance profile of the *L. monocytogenes* strains; this information is not available for this type of food. It is relevant that this research was authorized by the health authority. Therefore, the results in terms of the prevalence of *L. monocytogenes* in the food groups will improve targeted sampling of at-risk foods that promote the growth of this pathogen, as classified by the Chilean Sanitary Food Regulations.

## 5. Conclusions

The sampled foods did not comply with the microbiological criteria according to the Chilean Food Sanitary Regulations for *Listeria monocytogenes*. Higher than permissible counts, presence of virulence factors, serotypes associated with cases, and antibiotic-resistant strains mean that consuming moderately and minimally processed ready-to-eat foods is a tangible risk of disease for the particularly hypersensitive population that consumes them. In addition, more active food control and surveillance must be established by the health authorities. Education programs on preventive measures are needed for population groups at higher risk of listeriosis.

## Figures and Tables

**Figure 1 microorganisms-08-01669-f001:**
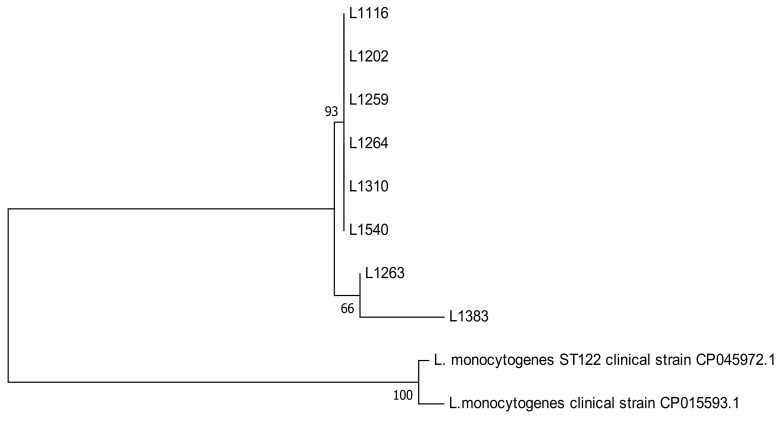
Phylogenetic tree of *hlyA* sequencing of *Listeria monocytogenes* identified in the study and two *L. monocytogenes* reference strains of clinical origin. The tree with the highest log likelihood (−469.20) is shown. The initial tree for the heuristic search was automatically obtained by applying Neighbor–Join and BioNJ algorithms to a matrix of pairwise distances estimated using the Maximum Composite Likelihood (MCL) approach. Evolutionary analyses were conducted with MEGA7 [35].

**Figure 2 microorganisms-08-01669-f002:**
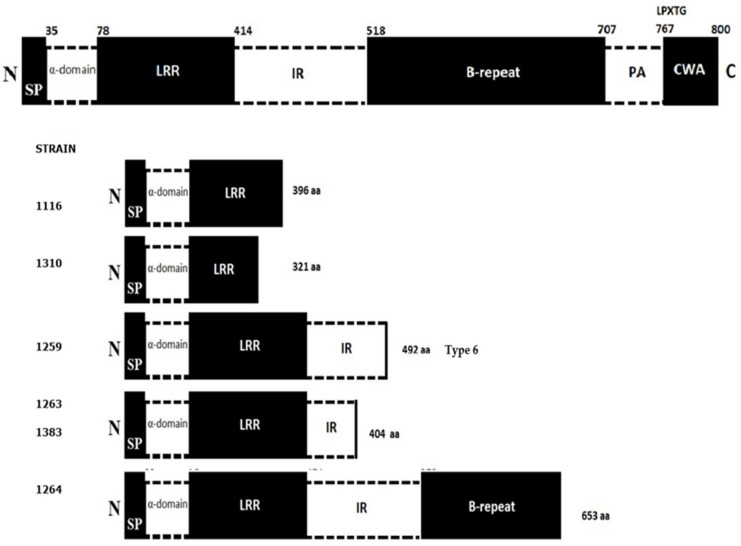
Full-length InlA and locations of premature stop codons (PMSC) in the *inlA* gene in the study. Full-length InlA represents the sequence for *Listeria monocytogenes* strain EGD-e. N: N-terminal end; S: signal sequence; LRR: leucine-rich repeat; IR: intergenic repeat; C: C-terminal end. Numbers below the EGD-e InlA sequence represent aa positions. Numbers at the right end represent the aa positions of the respective PMSC.

**Figure 3 microorganisms-08-01669-f003:**
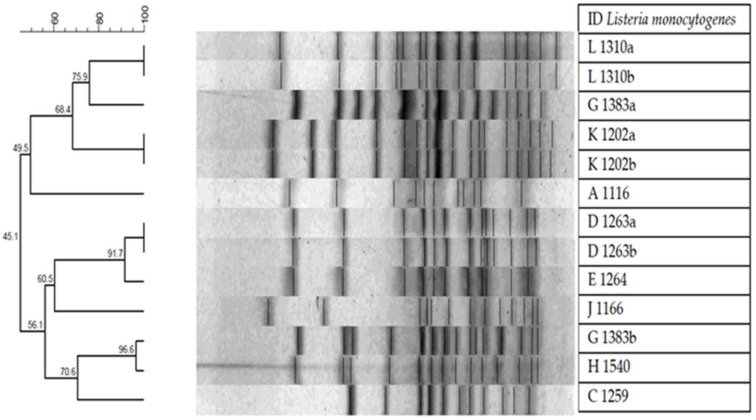
Pulsed-field gel electrophoresis (PFGE) of *Listeria monocytogenes* selected in the study. The test determined eight different groups of which four had more than two isolates. Strains 1202, 1263, and 1310 showed an identical pulsetype. Although from different food sources, strains 1310 and 1540 were very similar.

**Figure 4 microorganisms-08-01669-f004:**
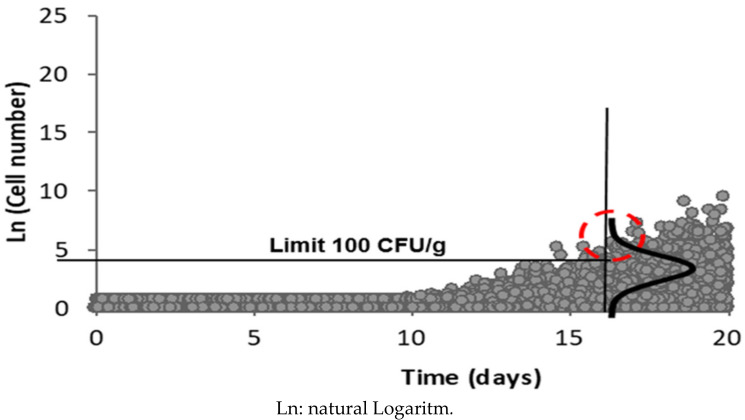
Estimation of the non-compliance of the regulation limit of 100 CFU/g established in the Chilean Food Sanitary Regulations. A total of 39% of ready-to-eat cooked meat products can exceed the *Listeria monocytogenes* limits at the end of their 16-day shelf-life.

**Table 1 microorganisms-08-01669-t001:** Primers used in this study.

Gene	Sequence	Amplification Product	Reference	Annealing (°C)
*hlyA*	f: 5′-ACTTCGGCGCAATCAGTGA-3′ r: 5′-TTGCAACTGCTCTTTAGTAACAGCTT-3′	137 bp	Zhang et al. [19]	60 °C
*hlyA*	f: 5′-CCTAAGACGCCAATCGAA-3′ r: 5′- AGCGCTTGCAACTGCTC	702 bp	Border et al. [23]	50 °C
*prfA*	f 5′-CGGGATAAAACCAAAACAATTT-3′ r: 5′-TGAGCTATGTGCGATGCCACTT-3′	508 bp	Klein y Juneja [24]	60 °C
*inlA*	f: 5′-GGCTGGGCATAACCAAATTA-3′ r: 5′-CTTTTGTTGGTGCCGTAGGT-3	629 bp	Montero et al. [25]	60 °C

**Table 2 microorganisms-08-01669-t002:** Positivity of *Listeria monocytogenes* in minimally and moderately processed artisanal foods.

Type of Processing	n	Positives		*p*-Value *
		n	(%)	
Minimally processed artisanal	95	11	(11.6)	0.0028
Moderately processed artisanal	305	19	(6.2)	
**Total**	**400**	**30**	**7.5**	

* Mann–Whitney test.

**Table 3 microorganisms-08-01669-t003:** Characteristics of strains isolated in food according to genotype, type of food, type of process, API profile, serotype, sequence type (ST), premature stop codons (PMSC), and antibiotic resistance profile.

Strain Number	Pulsotype	Type of Food	Type of Process *	Date	Count (CFU/g)	API Profile	Serotype	ST	Position of PMSC in inlA Gene	Antibiotic Resistance Profile ****					
										AMP (10 µg)	PEN (10 µg)	SXT (25 µg)	ERY (15 µg)	VAN (30 µg)	CHL (30 µg)
1116	A	Cooked meats	M	2017	1.6 × 10^2^	6510	1/2a	8	396	I	S	S	S	S	S
1202	K	Cooked meats	MD	2018	1.7 × 10^2^	6510	1/2a	8	NM ***	S	S	S	S	S	S
1259	C	Pre-processed fruit	M	2017	3.2 × 10^2^	6510	1/2a	ND **	492 Type 6	S	S	S	S	S	S
1263	D	Prepared meals and dishes	MD	2017	2.1 × 10^2^	6510	1/2a	8	404	S	S	S	S	S	S
1264	E	Prepared meals and dishes	MD	2017	1.2 × 10^2^	6510	1/2a	8	653	R	S	R	S	S	S
1310	L	Pre-processed fruit	M	2018	2.1 × 10^2^	6510	1/2a	8	321	I	S	S	S	S	S
1383	G	Cooked meats	M	2017	4.2 × 10^2^	6410	4b	388	404	R	S	S	S	S	S
1540	H	Cooked meats	MD	2017	4.4 × 10^2^	6510	1/2a	8	NM **	S	S	S	S	S	S

* M: Minimal processing; MD: Moderate processing. ** ND: Potentially a new ST. *** NM: No nutations associated with PMSC. **** Ampicillin (AMP), penicillin (PEN), trimethoprim/sulfamethoxazole (SXT), erythromycin (ERY), Vancomycin (VAN), chloramphenicol (CHL).

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
