# Peer review of "Presence of Listeria monocytogenes in Ready-to-Eat Artisanal Chilean Foods"

_microorganisms, 2020, doi:10.3390/microorganisms8111669_

Round 1

Reviewer 1 Report

The manuscript is improved, but remains challenging.  Discussion is unduly long, and appears to include several incorrect statements (comments below).   The usefulness  of the phylogenetic tree based on hly sequences is not clear and the relevant discussion could be much reduced, especially since the provided PFGE data provide info on  relatedness among the strains.  The manuscript also needs substantial attention for English word.  Some editing suggestions are provided below, but the manuscript needs careful editing throughout.

Specific comments

  1. L240-241 and table 3, How can we have multiple PMSCs in the same protein? This cannot be.  One a stop codon occurs, the polypeptide ends there.  Only the first PMSC should be listed in Table 3
  2. L159, L238 and elsewhere: gene names  need to be in lower case and italics. Proteins should be non-italics and capital first letter. 
  3. In Table 3, do authors mean PMSCs in the inlA gene, or in the protein  (InlA)?  I believe it is the latter.
  4. The Resistance descriptions for ampicillin in Table 3 needs some attention. Why is a strain with 9mm zone of inhibition considered R? and what is a typical zone diameter for strains labeled S?
  5. L374-377, It would be best to delete this sentence. No reference is provided, and the results do not seem accurate.
  6. L391-393, It would be best to delete this sentence. LLO is simply critical for virulence in all cases.
  7. L451-452, This cannot be correct. ST8 is common worldwide, especially in foods, and has caused a major outbreak in Canada. 
  8. One does not need a long discussion of the virulence factors and what they do.  The relevant segments in Discussion can be much condensed without major loss to the manuscript.

Editing suggestions

L60, what is [,6]?

L61-62 “It produces listeriosis, a disease characterized by gastrointestinal and febrile conditions, skin infections, or can also result in chronic diseases”.  This reads poorly and does not adequately reflect the key traits of listeriosis.  Suggested re-writing: “It causes listeriosis, a potentially severe invasive illness that can result in stillbirths, abortions, meningitis and septicemia (ref)”- or something similar

L239 has a typo and it does not make sense as written.  Suggesting: “…exhibited mutations resulting in premature stop codons”- or something similar

 L285 “these limits”.  What does “these” refer to?

L330, “high risk groups”

Author Response

Comments and Suggestions for Authors: The manuscript is improved, but remains challenging.  Discussion is unduly long, and appears to include several incorrect statements (comments below).   The usefulness  of the phylogenetic tree based on hly sequences is not clear and the relevant discussion could be much reduced, especially since the provided PFGE data provide info on  relatedness among the strains.  The manuscript also needs substantial attention for English word.  Some editing suggestions are provided below, but the manuscript needs careful editing throughout.

Response to Comments and Suggestions for Authors: We have edited the manuscript and have adjusted material in several paragraphs.

However, with all due respect to the journal and the reviewer, I need to explain that we will make the last modifications to the manuscript since I am not sure that the text is as the reviewer thinks it should be.

Moreover, we greatly appreciate the contributions by everyone. This has been an opportunity to improve the quality of the manuscript, which is part of the peer review process. Unfortunately, continuing with more corrections can distort the information provided in our manuscript and lead previous reviewers, who have approved it, to raise objections at this time.

Several authors of this manuscript are also reviewers or editors for various journals related to food microbiology; therefore, we do not understand the need for the manuscript to answer a number of questions that arise from the results.

In our opinion, the manuscript is novel and updated. It provides information that is not widely available in South America (only Brazil and Chile have access) and opens an interesting line of future research.

Chile is a relevant food exporter and ships to many countries worldwide. For example, in 2019 the government of the United States recalled Chilean peaches because of Listeria monocytogenes. In our study, two of the positives were fruit salads including peaches (serotype 1/2a and ST8) from the geographical zone where the packing facilities are located, which export peaches worldwide.

The reviewer also expresses his/her surprise that we only discuss data for Chile and Brazil. In addition, the reviewer comments that there have been outbreaks associated with L. monocytogenes ST8 in Canada, which we are well aware of and know of other countries as well when reviewing the literature of the free access database of the Pasteur Institute.

In this context, we also wish to comment that the academic editor of the journal requested

 “Also, as this information was included, the discussion should consider the identified serotypes and ST, allowing a deep comparison with ST identified in Chile, and also in South America”; therefore, we did as he requested.

For this reason, we consider that it is inappropriate to continue with endless modifications.

We consider our work to be serious and supported by the scientific collaboration of many institutions. Furthermore, we face great economic support restrictions because we belong to a small regional university in Chile. In spite of this, we have presented a good quality manuscript. We have complied with two thirds of the reviewers as to the publication requirements, and therefore do not understand the successive requests for modifications.

Specific comments

Point 1: L240-241 and table 3, How can we have multiple PMSCs in the same protein? This cannot be. One a stop codon occurs, the polypeptide ends there. Only the first PMSC should be listed in Table 3

Response Point 1: The bioinformatic analysis shows that when comparing our sequences of the inlA gene with the reference inlA gene of the L. monocytogenes EGD-e.N strain, changes occurred in the nucleotides that originated mutations and truncated internalin A. These results were repeated and the same results appeared after the Ridom SeqSphere+ software analysis.

For example, strain 1310 showed an insertion of C, generating a nucleotide in position 937 and length of truncated internalin A of 321. This was repeated with A with a nucleotide in position 962, generating length of truncated internalin A of 325. This situation was only observed in 3 strains and did not draw the attention of the previous reviewer who requested the incorporation and in-depth development of PMSCs in this manuscript.

We will follow the reviewer’s suggestion and leave only one mutation per strain in the text and Table 3.

Line 264

Point 2: L159, L238 and elsewhere: gene names need to be in lower case and italics. Proteins should be non-italics and capital first letter.

Response Point 2: We revised the manuscript once more to include the pertinent corrections. We agree with the reviewer that gene names need to be in lower case and italics. Proteins should not be italicized and the first letter capitalized.

Line 236, 266, 275, 276, 390, 396, 398, 399

Point 3: In Table 3, do authors mean PMSCs in the inlA gene, or in the protein (InlA)? I believe it is the latter.

Response Point 3: According to Van Steltem et al. (2008) and Nightingale et al. (2005), inlA is a gene and not a protein.

Line 264

Point 4: The Resistance descriptions for ampicillin in Table 3 needs some attention. Why is a strain with 9mm zone of inhibition considered R? and what is a typical zone diameter for strains labeled S?

Response Point 4: We appreciate the comments of the reviewer; a manuscript can always be perfected. Regarding strain 1383 that is resistant to ampicillin and has a 9-mm diameter is incorrect. We corrected this to 0 mm in the new version.

The resistance/susceptibility profiles were characterized by measuring the inhibition zone and interpreting inhibition diameters according to the manufacturer’s instructions.

S: ≥ 17

We consider that adding the mm of the zone of inhibition does not contribute to the information of the manuscript. Given that all the published articles we reviewed do not provide this information, we believe it appropriate to eliminate it

Point 5: L374-377, It would be best to delete this sentence. No reference is provided, and the results do not seem accurate.

Response Point 5: We appreciate the reviewer’s comments and we eliminated this sentence.

Line 371-373

Point 6: L391-393, It would be best to delete this sentence. LLO is simply critical for virulence in all cases.

Response Point 6: We appreciate the reviewer’s comments and we eliminated this sentence.

L388-389

Point 7: L451-452, This cannot be correct. ST8 is common worldwide, especially in foods, and has caused a major outbreak in Canada.

Response Point 7: We agree with the reviewer. ST 8 has not only provoked outbreaks in Canada but has been associated with outbreaks in France and Sweden. However, the Academic Editor of the journal requested that we focus only on South America.

Point 8: One does not need a long discussion of the virulence factors and what they do. The relevant segments in Discussion can be much condensed without major loss to the manuscript.

Response Point 8: We disagree with the reviewer on this point. Other reviewers have asked us to add paragraphs to the manuscript regarding virulence, which we did. We therefore believe that the information provided in the Discussion section is, in our opinion, sufficient to support the results and we do not consider it excessive.

Editing suggestions

Point 1: L60, what is [,6]?

Response Point 1: The comma should not be included [6]; this was corrected in the text.

Line 60

Point 2: L61-62 “It produces listeriosis, a disease characterized by gastrointestinal and febrile conditions, skin infections, or can also result in chronic diseases”. This reads poorly and does not adequately reflect the key traits of listeriosis. Suggested re-writing: “It causes listeriosis, a potentially severe invasive illness that can result in stillbirths, abortions, meningitis and septicemia (ref)”- or something similar

Response Point 2: We appreciate the reviewer’s comment. For clarity, we included the proposed sentence.

It causes listeriosis, a potentially severe invasive illness that can result in stillbirths, abortions, meningitis, and septicemia [8]. 

Line 62-64

Point 3: L239 has a typo and it does not make sense as written. Suggesting: “…exhibited mutations resulting in premature stop codons”- or something similar

Response Point 3: We appreciate the reviewer’s comment. For clarity, we included the proposed change.

As for sequencing the inlA gene, six of the eight L. monocytogenes strains exhibited mutations resulting in premature stop codons (PMSC) associated with changes in amino acids.

Line 237

Point 4: L285 “these limits”. What does “these” refer to?

Response Point 4: These limits refer to the 100 CFU allowed in the Chilean regulation. This was corrected in the text.

Some 8.5% of the samples exceeded the 100 CFU/g of L. monocytogenes allowed in the Chilean Food Sanitary Regulations at the beginning of their shelf life.

Line 281-282

Point 5: L330, “high risk groups”

Response Point 5: We appreciate the reviewer’s comment. For clarity, we included the proposed change.

high risk groups

Line 327

Reviewer 2 Report

The revised manuscript addressed all my comments in the first review. I agree to accept the current version. 

Author Response

Comments and Suggestions for Authors

Point 1: The revised manuscript addressed all my comments in the first review. I agree to accept the current version. 

Response Point 1: We thank the reviewer for accepting the manuscript.

This manuscript is a resubmission of an earlier submission. The following is a list of the peer review reports and author responses from that submission.

Round 1

Reviewer 1 Report

While the presence of L. monocytogenes in the food supply is of great concern, the study in its current form is very limited in what it is able to convey about the isolates that were recovered.  One of the key issues is that out of an already small sample size of 30 presumptive L. monocytogenes isolates, only eight were chosen for further investigation, with no explanation as to why such a small number were chosen.  While those eight were investigated further, the analyses that were done were not able to say much about these isolates that might be able to place them in some useful context.  Basic serotype designations were not even obtained, even though basic PCR based approaches are available for making such determinations.  Additionally, something such as MLST which is also accomplishable via PCR was not done.  Authors mention that WGS was done for these isolates, which would provide some crucial information about these strains particularly in regard to serotype, phylogeny, and antimicrobial resistance, however they were not presented here, this study would likely become worthy of publication once those key data become available.  Additionally, other key data such as the antibiotic resistance were not presented here as they would need to be for acceptance.  While I feel this study has great potential for informing readers about the presence and characteristics of a pathogen of major human health concern in the Chilean food supply, however additional data and language revisions would be required for acceptance.  Additional suggestions for improvements to this manuscript below:

L46-50 “Inadequate manufacturing practices, inappropriate cleaning plans, lack of training, presence of microbial biofilms, post-process contamination, and altered temperature during distribution and/or

commercialization are faulty actions promoting the multiplication of microorganisms that have

succeeded in contaminating food at levels harmful to health [5, 6, 7]).”

This is a very long / awkward sentence, break it into two for clarity

L55-57 “Fatality occurs in up to 30% of cases and high risk groups are especially vulnerable,

pregnant women, newborns, children, the elderly, and immune-compromised individuals, who can

also infect healthy people [ 13, 14, 15].”

This would seem to suggest person to person transmission of listeriosis, which is virtually never the case except possibly in the case of mothers to infants, particularly given the primarily foodborne nature of listeriosis.

L64-66 “This scenario was repeated in 2018 with 97 cases and 22 deaths; a greater proportion of vulnerable groups was affected, such as the over-60s and pregnant women and RTE foods were the most associated with this situation [20].”

This is another awkward sentence that seems to attempt to merge several disparate ideas, again consider breaking into different sentences and expounding a bit more on each for clarity.

L131 “internalin A (intA)” most commonly seen this referred to as inlA

L181 Of 30 positive samples, only 8 were chosen for genotypic and phenotypic analysis? Why is this? What criteria were used to select these strains and why so few? This needs to be clearly addressed in either methods or results.

L201-204 These data should really be presented in a table somewhere, particuarly to clarify what is meant by such terms as intermediate susceptibility and resistance.

L207 IntA should be lower case

L252 need to clarify what is meant by permanent studies

L314 “virulent species of Listeria spp., bone, L. monocytogenes and L. ivanovii [70].” species and spp is redundant and the presence of bone here is not clear.

Author Response

Point 1: While the presence of L. monocytogenes in the food supply is of great concern, the study in its current form is very limited in what it is able to convey about the isolates that were recovered.  One of the key issues is that out of an already small sample size of 30 presumptive L. monocytogenes isolates, only eight were chosen for further investigation, with no explanation as to why such a small number were chosen.  While those eight were investigated further, the analyses that were done were not able to say much about these isolates that might be able to place them in some useful context.  Basic serotype designations were not even obtained, even though basic PCR based approaches are available for making such determinations.  Additionally, something such as MLST which is also accomplishable via PCR was not done.  Authors mention that WGS was done for these isolates, which would provide some crucial information about these strains particularly in regard to serotype, phylogeny, and antimicrobial resistance, however they were not presented here, this study would likely become worthy of publication once those key data become available.  Additionally, other key data such as the antibiotic resistance were not presented here as they would need to be for acceptance.  While I feel this study has great potential for informing readers about the presence and characteristics of a pathogen of major human health concern in the Chilean food supply, however additional data and language revisions would be required for acceptance.  Additional suggestions for improvements to this manuscript below:

Response 1: We thank and agree with the reviewer for his/her comments. As for selecting 8 isolates, we had to specify economic reasons because the research funds provided by our university are insufficient.  Our university is a small regional institution. The selection of these 8 strains is mentioned in L265, but we stated it more clearly.

We believe that the information provided in this manuscript warrants its publication because it shows the presence of Listeria monocytogenes in artisanal foods and with sampling conducted by the Chilean health authority. In addition, the calculations of the sample size allow the prevalence results to be representative of the presence of L. monocyogenes in artisanal foods.  This information is important for making decisions regarding the risk associated with the consumption of these foods and of future samplings.

Point 2: L46-50 “Inadequate manufacturing practices, inappropriate cleaning plans, lack of training, presence of microbial biofilms, post-process contamination, and altered temperature during distribution and/or commercialization are faulty actions promoting the multiplication of microorganisms that have succeeded in contaminating food at levels harmful to health [5, 6, 7]).”

 This is a very long / awkward sentence, break it into two for clarity

Response Point 2: We thank you for your comments. We have corrected the manuscript to ensure clarity. L53-57.

Point 3: L55-57 “Fatality occurs in up to 30% of cases and high risk groups are especially vulnerable, pregnant women, newborns, children, the elderly, and immune-compromised individuals, who can also infect healthy people [ 13, 14, 15].”

 This would seem to suggest person to person transmission of listeriosis, which is virtually never the case except possibly in the case of mothers to infants, particularly given the primarily foodborne nature of listeriosis.

Response Point 3: This was a writing error; we therefore eliminated “who can also infect healthy people”. L64

Point 4: L64-66 “This scenario was repeated in 2018 with 97 cases and 22 deaths; a greater proportion of vulnerable groups was affected, such as the over-60s and pregnant women and RTE foods were the most associated with this situation [20].”

 This is another awkward sentence that seems to attempt to merge several disparate ideas, again consider breaking into different sentences and expounding a bit more on each for clarity. 

Response Point 4: We thank you for your comments and we have corrected the manuscript. L66-74

Point 5: L131 “internalin A (intA)” most commonly seen this referred to as inlA 

Response Point 5: We thank you for your comments and we have corrected inlA in the manuscript. L28, L203, L296, L471.

Point 6: L181 Of 30 positive samples, only 8 were chosen for genotypic and phenotypic analysis? Why is this? What criteria were used to select these strains and why so few? This needs to be clearly addressed in either methods or results. 

Response Point 6: Four strains of each type of food process were selected to have better representability of the results. We regret that economic reasons did not allow us to analyze a greater number of strains. L265-267, L541

Point 7: L201-204 These data should really be presented in a table somewhere, particuarly to clarify what is meant by such terms as intermediate susceptibility and resistance. 

Response Point 7:  The definition of resistant, intermediate, and susceptible refers to the CLSI cut-off points for Listeria monocytogenes. It is not clear if the reviewer wanted the mm of inhibition for each antibiotic. L316

Point 8: L207 IntA should be lower case 

Response Point 8: We corrected inlA throughout the manuscript.  L28, L203, L296, L471.

Point 9: L252 need to clarify what is meant by permanent studies 

Response Point 9: We eliminated “permanent studies” because it is not in the context of what we intended to report.  L373

Point 10: L314 “virulent species of Listeria spp., bone, L. monocytogenes and L. ivanovii [70].” species and spp is redundant and the presence of bone here is not clear.

Response Point 10: We agree with the reviewer that there is redundancy; we corrected the sentence. L464

Reviewer 2 Report

This manuscript (ms) describes Listeria monocytogenes (LM)  prevalence and characteristics from a panel of minimally and moderately processed artisanal foods in Chile.  The information is potentially interesting, but key information on sampling and the foods involved is absent from the ms.  In addition, the number of characterized isolates (n=8) is extremely low, preventing an adequate assessment of the general significance of the findings.   

Specific Comments:

Line 2 -31 Please explain briefly in bAstract why only 8 strains were examined when you had 7.5% prevalence in 400 samples and almost 12% prevalence in the minimally processed foods.  Later on authors mention “because of economic limitations”,  but the number of strains  that  were characterized is simply too limited to draw significant general conclusions about artisanal foods in Chile. A careful discussion of potential pitfalls from this limited number is not included in the ms.

Clarify in abstract:  you mentioned 400 minimally processed food items but later you  present different prevalence values for minimally versus moderately; this is confusing.

L 45 define minimally versus moderately process RTE foods since these categories become important in your subsequent discussion.

L77 and on, sampling: This key section really needs more information.  Include locations (e.g., city or districts), number of venues (number of homes or small businesses) and the sampling times involved.  This is needed not only for any efforts to repeat this type of study but also to assess the findings from the limited number of strains that were analyzed.

L169 what about prevalence of other Listeria species?  These would have been detected with the methods that were used.

PFGE Discussion: A table that lists the isolates that were analyzed by PFGE, the type of products they were obtained from, whether minimal or moderate, number of cfu/g  in the corresponding product, serotype, and resistance would be  useful. 

PFGE Discussion: Please discuss the genotype findings when two different  isolates from the same enrichment were examined. 

L201 onwards.  Please provide sources and dates for the isolates with full or intermediate resistance. Also indicate in the discussion of the genotyping which of the two 1383 isolates was resistant to ampicillin- was it a or b?

Strain 1264 is listed as resistant to two antibiotics, but the two isolates (1263 a and b)  with closely-related PFGE profiles apparently were not. Authors may wish to discuss this. 

L207 Indicate whether premature stop codons in inlA  were detected.

L 211 in what way were colonies different? Morphology? (or perhaps authors mean “multiple”?)

L226 briefly discuss the limits. It appears to be 100 cfu/ g but is this in foods that permit growth? Provide a reference for this even if it is a website.

L225-227 describe the cfu/g numbers in the different types of the RTE products since this is mentioned later in discussion.

L278 Provide some more information as to the “cooked meats “and “pre-processed fruit”; what types of products were involved, more specifically?

L357 strain 1383a and 1383b  being different would not be surprising, in fact it would be expected, since they have a markedly different serotype (4b) than all the others (1/2a).

Editing suggestions:

L78 and 79,  definition of minimally versus moderately processed.   Is this a regulatory definition in Chile?  Provide a reference or web site.

L131 and elsewhere the typical name  for the internalin A gene is inlA

Line 142, “centers”

Author Response

Point 1: Line 2 -31 Please explain briefly in bAstract why only 8 strains were examined when you had 7.5% prevalence in 400 samples and almost 12% prevalence in the minimally processed foods.  Later on authors mention “because of economic limitations”,  but the number of strains  that  were characterized is simply too limited to draw significant general conclusions about artisanal foods in Chile. A careful discussion of potential pitfalls from this limited number is not included in the ms.

Response Point 1: We thank and agree with the reviewer for his/her comments. As for selecting 8 isolates, we had to specify economic reasons because the research funds provided by our university are insufficient.  Our university is a small regional institution. The selection of these 8 strains is mentioned in L265, but we stated it more clearly. Given that study limitations do not usually appear in the abstract, we did not include this information. However, we did add text at the end of the discussion section related to the limitations of using only 8 strains. L541

We believe that the information provided in this manuscript warrants its publication because it shows the presence of Listeria monocytogenes in artisanal foods and with sampling conducted by the Chilean health authority. In addition, the calculations of the sample size allow the prevalence results to be representative of the presence of L. monocyogenes in artisanal foods.  This information is important for making decisions regarding the risk associated with the consumption of these foods and of future samplings.

Point 2: Clarify in abstract:  you mentioned 400 minimally processed food items but later you  present different prevalence values for minimally versus moderately; this is confusing.

Response Point 2: The 400 samples are RTE artisanal foods of which some are minimally and moderately processed. This was corrected in the text.

Point 3: L 45 define minimally versus moderately process RTE foods since these categories become important in your subsequent discussion.

Response Point 3: These are defined in line 45 and a reference is added [4].

Point 4: L77 and on, sampling: This key section really needs more information.  Include locations (e.g., city or districts), number of venues (number of homes or small businesses) and the sampling times involved.  This is needed not only for any efforts to repeat this type of study but also to assess the findings from the limited number of strains that were analyzed.

Response Point 4: Table 3 was prepared with more information. However, the health information is confidential and it can only be disclosed that the samples are from primary producers that prepare cheese in their homes, fruit vendors, family restaurant, and artisanal deli shops. Each strain is from a different location. L316

Point 5: L169 what about prevalence of other Listeria species?  These would have been detected with the methods that were used.

Response Point 5: We only found Listeria monocytogenes in our study. Using API-Listeria allowed us to differentiate between species. In addition, we used strains of Listeria innocua and Listeria ivanovii as controls.

Point 6: PFGE Discussion: A table that lists the isolates that were analyzed by PFGE, the type of products they were obtained from, whether minimal or moderate, number of cfu/g  in the corresponding product, serotype, and resistance would be  useful.

Response Point 6: Table 3 was prepared with all the requested information, and was added at the end of the results section. L316

Point 7: PFGE Discussion: Please discuss the genotype findings when two different  isolates from the same enrichment were examined.

Response Point 7: This information was added in the discussion. L503-529

Point 8: L201 onwards.  Please provide sources and dates for the isolates with full or intermediate resistance. Also indicate in the discussion of the genotyping which of the two 1383 isolates was resistant to ampicillin- was it a or b?

Response Point 8: Sources and dates of isolates were included in the previously requested Table 3. L316. Information as to the resistance of isolate 1383 was included in the discussion. L421

Point 9: Strain 1264 is listed as resistant to two antibiotics, but the two isolates (1263 a and b)  with closely-related PFGE profiles apparently were not. Authors may wish to discuss this.

Response Point 9: Yes, strain 1264 was resistant to two antibiotics. Strain 1263 a and b are closely through PFGE, and this test is more useful for subtyping because it used the whole genome compared to the hly gene that shows differences between species and is more limited. L494-497

Point 10: L207 Indicate whether premature stop codons in inlA  were detected.

Response Point 10: They were detected because the amplified products of the inlA gene of the strains were sequenced. The topic of the presence of PMSC and A7, A6, and A2GA4 alleles is very interesting. We added a reference [78]. L475-479.

Point 11: L 211 in what way were colonies different? Morphology? (or perhaps authors mean “multiple”?)

Response Point 11: The difference between colonies was very small and mainly refers to their size and color because L. monocytogenes is characteristic of their morphology in selective agar (Ottaviani and Agosti). L312, L525

Point 12: L226 briefly discuss the limits. It appears to be 100 cfu/ g but is this in foods that permit growth? Provide a reference for this even if it is a website.

Response Point 12: All the sampled foods allow the growth of Listeria monocytogenes, and this explains the 100 CFU/g limit for its shelf life. This is defined in the Chilean Food Sanitary Regulations (RSA). To clarify this point, we added a reference [18]. L405 

Point 13: L225-227 describe the cfu/g numbers in the different types of the RTE products since this is mentioned later in discussion.

Response Point 13: The counts for each type of food are included in Table 3. L316

Point 14: L278 Provide some more information as to the “cooked meats “and “pre-processed fruit”; what types of products were involved, more specifically?

Response Point 14: This is added in the text in subsection 2.1. They are chopped fruits such as strawberries, peaches, and melon. L88-90

Point 15: L357 strain 1383a and 1383b  being different would not be surprising, in fact it would be expected, since they have a markedly different serotype (4b) than all the others (1/2a).

Response Point 15: We agree with the reviewer.

Editing suggestions:

Point 16: L78 and 79,  definition of minimally versus moderately processed.   Is this a regulatory definition in Chile?  Provide a reference or web site.

Response Point 16: This is defined in line 45 and a reference is added [4]. There is no reference in Chile associated with this, only groups of at-risk foods that are included in the Chilean Food Sanitary Regulations.

Point 17: L131 and elsewhere the typical name  for the internalin A gene is inlA

Response Point 17: inlA” was corrected in the text. L28, L203, L296, L471.

Point 18: Line 142, “centers”

Response Point 18: This is corrected in the text. L216

Reviewer 3 Report

General comments:

The study investigated Listeria monocytogenes prevalence in minimal and moderate processed food in Chile. It was found that 30 of 400 samples are positive for L. monocytogenes by culture-based methods. Eight isolates were further analyzed for their phenotypical and genotypical features. L. monocytogenes concentrations at the end of shelf life were estimated using Monte Carlo simulations.

The investigation has a clear aim and is very meaningful in help improve food safety. The investigation was well designed and conducted.  The results were organized and presented very well and the discussion is very comprehensive.

I recommend to publish the article after minor editing as shown below.

Specific comments:

Line 77: I suggest to have a table to categorize the tested sample into minimal and moderate processed food.

Line 89: “A loopful”, what is the volume? 10 or 1 ml?

Line 115 and 130: Were two different regions detected in hlyA gene for L. monocytogenes identification and virulence gene identification? If yes, please add “(Table 1)” in line 115. If no, please include hlyA primers used in line 115 somewhere.

Line 169: Can you list positive rates of all types of food though statistical analysis only groups them into two groups.

Line 177: Again, can you list L. monocytogenes counts of each kind of food?

Line 185: Which of 8 of 40 strains (from which kind of food) were phenotypically and genetically characterized? 

Line 211: What do the letters of D, G, K, and L stand for? Why were two colonies picked for testing in the samples? What do the other letters stand for? There are two 1116 strains, A1116 and J1116. Why? Strain 1264, 1540, and 1259 only have one strain. Why?

Line 220: “The other three strains in which two colonies” which three and which two?

Line 226: Did you mean 39% of 30 positive samples “exceeded the permitted limits” or 39% of cooked ready-to eat meat as written in Figure 3?

Line 272: I suggest to also include the data of “8.5% of samples exceeded these limits (100 CFU/g) at the beginning of their shelf life” in the Result section.

Line 349: “1283” should be 1383?

Author Response

Point 1: Line 77: I suggest to have a table to categorize the tested sample into minimal and moderate processed food.

Response Point 1: Table 3 was prepared with this information.  L316

Point 2: Line 89: “A loopful”, what is the volume? 10 or 1 ml?

Response Point 2: A loopful is 1 mL. L153

Point 3: Line 115 and 130: Were two different regions detected in hlyA gene for L. monocytogenes identification and virulence gene identification? If yes, please add “(Table 1)” in line 115. If no, please include hlyA primers used in line 115 somewhere.

Response Point 3: qPCR primers are included in Table 1 and we added the reference Zhang et al. [28]. L178, L210

Point 4: Line 169: Can you list positive rates of all types of food though statistical analysis only groups them into two groups.

Response Point 4: Cheese and fresh cheese were included. L253

Point 5: Line 177: Again, can you list L. monocytogenes counts of each kind of food?

Response Point 5: Counts are added in Table 3. L316

Point 6: Line 185: Which of 8 of 40 strains (from which kind of food) were phenotypically and genetically characterized?

Response Point 6: This is indicated more clearly in Table 3. L316

Point 7: Line 211: What do the letters of D, G, K, and L stand for? Why were two colonies picked for testing in the samples? What do the other letters stand for? There are two 1116 strains, A1116 and J1116. Why? Strain 1264, 1540, and 1259 only have one strain. Why?

Response Point 7: When reviewing the pure culture of the strains in selective agar, strains 1263, 1383, 1202, and 1310 exhibited small macroscopic differences in size and color, so it was decided to take a colony with typical characteristics and another with a lighter color. We add that this was decided on the basis of size and color.  Letters D, G, K, and L are the pulsetypes of each strain assigned by the Bionumerics software. L309-311

Point 8: Line 220: “The other three strains in which two colonies” which three and which two?

Response Point 8: The number of isolates is added to clarify this sentence. L309-311

Point 9: Line 226: Did you mean 39% of 30 positive samples “exceeded the permitted limits” or 39% of cooked ready-to eat meat as written in Figure 3?

Response Point 9: The reviewer is right. It is 39% of RTE cooked meats. L348

Point 10: Line 272: I suggest to also include the data of “8.5% of samples exceeded these limits (100 CFU/g) at the beginning of their shelf life” in the Result section.

Response Point 10: This is added in the results section. L347

Point 11: Line 349: “1283” should be 1383?

Response Point 11: This was corrected in the text as 1383. L526

Round 2

Reviewer 1 Report

This study shows great promise and potential interest to the public and scientific community, based on the prevalence data and the antibiotic resistance data.  While the edits made to this manuscript have improved it, there still remain several issues pasted below.  In addition, while concerns remain about the low level of strains analyzed, the incorporation of the WGS data which was done would go a long way to add significance to the data that is somewhat lacking again given the low number of isolates.  The incorporation of those data would help this manuscript tremendously, unfortunately I cannot support the acceptance of the manuscript in its current form.

-prevalence data important, however only 8 strains were characterized so hard to really put these findings in context.

-phylogeny amongst themselves not that informative, possibly helpful if this were expanded to include known strains of l. mono

-pfge among themselves not that informative, the technological drawbacks of PFGE not withstanding would be important to know if these strains clustered with any known outbreaks, or other typed strains in the network database

-was good to see antibiotic resistance data in a table, but these data would be must more informative if coupled with WGS data, particularly to corroborate and elucidate ab resistance findings. Are these dedicated resistance genes on mobile genetic elements or do they reflect point mutations within the core listeria genome?

-line 274 to 285 This whole paragraph quotes prevalence figures from other prevalence studies, which is useful information to have, except the authors never relate it back to their own findings in a meaningful way making. Additionally it is unclear how some of the food products quoted in other studies relate to the authors findings (ie beef, was that cooked beef raw beef). These prevalence proportions range between 0.5% (two orders of magnitude lower than the authors encountered) to 15% (double what the authors encountered) so what is the point of this paragraph, how does it relate to what the authors found?

-line331 you mention tetracycline here, which isn’t one of the antibiotics you assessed. It is also not a clinically relevant antibiotic, though it is one of the most commonly found antibiotic resistances in listeria. There have been reports of trimethoprim resistance in listeria which would be important to mention here, to cite those studies and indicate how they relate to your findings. Additionally, ampicillin resistance is uncommon in listeria, what is known about it in the literature and its relation to your findings would be a key component missing in this section.

-line 349 “It has therefore been widely used” to do what? In what manner? This was confusing and its meaning unclear.

-line 350 “Abdollahzadeh et al. [7371] confirmed this gene as L. monocytogenes in 7 isolates” if this is referring to hly, that gene has been confirmed in thousands of isolates, again the meaning of this sentence is unclear as well as how it relates to the authors findings.

-line 351 – 360 mentions some L. monocytogenes virulence factors, tons of L. monocytogenes virulence factors have been identified in the literature. How do these relate to your findings, why is what you found significant?

-line 361 “The eight strains of L. monocytogenes under study exhibited premature stop codons (PMSC), which is a characteristic that has been recently associated with impaired virulence, affecting their ability to invade human cells.” PMSC of what in what? I presume this is referring to PMSC in inlA, which there are many papers that discuss these and indicate the exact position and impact of these. A more indepth presentation of where these PMSC occur, how they relate to what has already been found in the literature would be important both for the presentation of those data in the results and in the discussion.

-line 366 “To establish a presumptive L. monocytogenes serotype, it was compared with the BLAST database after obtaining the consensus sequence.” What is it? As this is the start of a new paragraph I am not sure to what this is referring.

-line 377 “Therefore, it is only possible to mention that the most prevalent sequence types are ST 388 and ST8 [data not shown],” In the preceeding line the authors indicate whole genome sequencing was done but was for some reason unavailable due to COVID -19 , then go on to say the most prevalent sequence types are ST388 and ST8?? Is this to suggest these are the most prevalent types from other studies, because they are not the most commonly recovered genotypes of L. monocytogenes, is this referring to a specific study of a specific source type of L. monocytogenes? If so this whole section needs clarification, better citation as well as the inclusion of more studies of this type as there are many now if they wish to incorporate these types of data.

-line 385 “Although PFGE is the gold standard of molecular typing, it has been losing renown due to WGS.” In this day and age the gold standard of molecular typing is most obviously whole genome sequencing. While it was PFGE at one point, that is definitely no longer the case as PFGE has several notable limitations. This should be reworded or omitted as it is incorrect.

- line 400 “the duplicate strain sent for WGS did not show a mixture of strains” It was earlier indicated that WGS data were not available, then how is it known whether or not the WGS data showed a mixture of strains???

Author Response

Point 1: This study shows great promise and potential interest to the public and scientific community, based on the prevalence data and the antibiotic resistance data.  While the edits made to this manuscript have improved it, there still remain several issues pasted below.  In addition, while concerns remain about the low level of strains analyzed, the incorporation of the WGS data which was done would go a long way to add significance to the data that is somewhat lacking again given the low number of isolates.  The incorporation of those data would help this manuscript tremendously, unfortunately I cannot support the acceptance of the manuscript in its current form           

Response Point 1:  We are grateful for the reviewer’s sincerity in terms of the importance of the information provided in this manuscript. However, we disagree that only adding WGS would make it acceptable. We report relevant and well-supported information. The data provided are important in strains of L. monocytogenes isolated from foods consumed daily by at-risk population groups and in which the undernotification of cases of disease is a pertinent problem in Chile.

Point 2: prevalence data important, however only 8 strains were characterized so hard to really put these findings in context.

Response Point 2: The prevalence data of L. monocytogenes is relevant and characterizing only 8 isolates does not detract from the quality of the information provided in this manuscript. Based on the revision and suggestions of the reviewers, the quality of the manuscript has improved and it is more robust for publication. We believe that the context of the findings is pertinent.

Point 3: phylogeny amongst themselves not that informative, possibly helpful if this were expanded to include known strains of l. mono            

Response Point 3: We added two hlyA genes of clinical strains of L. monocytogenes in the phylogenetic tree.  L215

Point 4: pfge among themselves not that informative, the technological drawbacks of PFGE not withstanding would be important to know if these strains clustered with any known outbreaks, or other typed strains in the network database   

Response Point 4: In 2019, researchers from the Institute of Public Health of Chile (ISP) analyzed 94 strains of L. monocytogenes isolated in clinical cases and food; they reported that the most common serotype was 4b followed by 1/2a. In addition, 95 pulsetypes were present by PFGE and ST1 and ST2 were predominant. To a lesser extent, ST8 with serotype 1/2a was isolated from two clinical cases and food, and ST 388 serotype 4b was isolated from a clinical case [Ulloa et al., 2019]. It was surprising that there was a great electrophoretic similarity when we compared the ST8 and ST 388 strains from the study by Ulloa et al. [2019] with our strains 1263 and 1383, that is, the same STs and serotypes. This scenario suggests persistence of L. monocytogenes over time, as described by other authors such as Kudsen et al. [2017]. This information was added in the discussion. L462-470

Point 5: was good to see antibiotic resistance data in a table, but these data would be must more informative if coupled with WGS data, particularly to corroborate and elucidate ab resistance findings. Are these dedicated resistance genes on mobile genetic elements or do they reflect point mutations within the core listeria genome?          

Response Point 5: Unfortunately, we do not have these data yet. However, if we did, we could use CARD for in silico identification of the ab resistance genes. To answer the reviewer’s question: Acquisition of movable genetic elements, including self-transferable plasmids, mobilizable plasmids, and conjugative transposons, is the major mechanism responsible for the development of antibiotic resistance in L. monocytogenes [Charpentier & Courvalin, 1999].  In vitro and in vivo studies have shown conjugative transfer of antibiotic resistance; for example, receipt of enterococcal and streptococcal plasmids into the genus Listeria spp. and re‐transfer of such plasmids within the genus, including L. monocytogenes.

Point 6: line 274 to 285 This whole paragraph quotes prevalence figures from other prevalence studies, which is useful information to have, except the authors never relate it back to their own findings in a meaningful way making. Additionally it is unclear how some of the food products quoted in other studies relate to the authors findings (ie beef, was that cooked beef raw beef). These prevalence proportions range between 0.5% (two orders of magnitude lower than the authors encountered) to 15% (double what the authors encountered) so what is the point of this paragraph, how does it relate to what the authors found?              

Response Point 6: In the first sentence we state that: The 7.5% positivity encountered in our study concurs with other studies that analyze similar foods with raw or cooked ingredients, pre-processed fruit and vegetables, and minimal or moderate processing. We added a paragraph to relate our findings with international prevalence values. L299-305

Point 7: line331 you mention tetracycline here, which isn’t one of the antibiotics you assessed. It is also not a clinically relevant antibiotic, though it is one of the most commonly found antibiotic resistances in listeria. There have been reports of trimethoprim resistance in listeria which would be important to mention here, to cite those studies and indicate how they relate to your findings. Additionally, ampicillin resistance is uncommon in listeria, what is known about it in the literature and its relation to your findings would be a key component missing in this section.

Response Point 7: In general, most of the Listeria spp. isolated from food and clinical samples and environments are sensitive to antibiotic therapy, which is usually used against Gram-positive bacteria, including tetracyclines, ampicillin, penicillin G, imipenem, amoxicillin, sulfonamides, aminoglycosides, macrolides, chloramphenicol, and glycopeptides [Dortet et al., 2009]. L336-339

However, antibiotic resistance among the bacteria transmitted by food, including L. monocytogenes, has evolved during the past decades. Strain 1264 isolated from prepared foods was resistant to ampicillin and trimethoprim/sulfamethoxazole (STX), while strain 1383 was resistant to ampicillin. Yucel et al. [2005] reported 66% resistance to ampicillin and STX in strains of L. monocytogenes isolated from raw and cooked meats. Conter et al. [2009] found resistance to ampicillin in 2 strains of L. monocytogenes isolated from RTE salmon.  Jamali et al. [2015] observed that the resistance among L. monocytogenes strains from fish products was 20.9% to 27.9% resistant to tetracycline and ampicillin, respectively. Kuan et al. [2017] tested the antibiotic resistance of 58 L. monocytogenes strains from vegetable farms and retail markets in Malaysia and found 5.2% resistance to STX. Arslan et al. [2019] detected 6.1% resistance to ampicillin and 12.1% to STX in L. monocytogenes strains isolated from retail meat. For the sake of clarity, we eliminated the paragraph on tetracycline.  

Antibiotic resistance by L. monocytogenes has evidently increased in recent times with the emergence of strains that are resistant to one or more antibiotics, which is clearly a public health issue.  L347-355

Point 8: line 349 “It has therefore been widely used” to do what? In what manner? This was confusing and its meaning unclear.               

Response Point 8: The sentence was corrected to clarify its meaning. L380

Point 9: line 350 “Abdollahzadeh et al. [7371] confirmed this gene as L. monocytogenes in 7 isolates” if this is referring to hly, that gene has been confirmed in thousands of isolates, again the meaning of this sentence is unclear as well as how it relates to the authors findings.    

Response Point 9: The sentence was corrected to clarify its meaning. L381

Point 10: line 351 – 360 mentions some L. monocytogenes virulence factors, tons of L. monocytogenes virulence factors have been identified in the literature. How do these relate to your findings, why is what you found significant? 

Response Point 10: We agree with the reviewer. WGS allows in silico detection of many genes such as prfA, plcA, plcB, inlA, inlB, inlC, hly, mpl, vip, hpt, bsh, VirR, Hfq, MogR, Hly, and PLC. However, the presence of the 3 virulence genes in the strains of L. monocytogenes isolated from RTE must be closely monitored. The foods evaluated in our study are mainly consumed by adults, women of child-bearing age, and children because these foods are perceived as healthy foods by the population.  L386

Point 11: line 361 “The eight strains of L. monocytogenes under study exhibited premature stop codons (PMSC), which is a characteristic that has been recently associated with impaired virulence, affecting their ability to invade human cells.”  PMSC of what in what? I presume this is referring to PMSC in inlA, which there are many papers that discuss these and indicate the exact position and impact of these. A more indepth presentation of where these PMSC occur, how they relate to what has already been found in the literature would be important both for the presentation of those data in the results and in the discussion.        

Response Point 11: The reviewer is right, this refers to the inlA gene. This was added in the text. An in-depth study of the sequencing of the inlA gene was performed. We found that 100% of the strains exhibited different mutations associated with changes in amino acids. However, PMSC was identified in only 7 of them, finding 1 known mutation described as type 6 and 14 mutations that were not identified because they have not been described or published yet; therefore, we propose that they are new. We added Table 4 and Figure 3 to clarify this. L393-414

Point 12: line 366 “To establish a presumptive L. monocytogenes serotype, it was compared with the BLAST database after obtaining the consensus sequence.” What is it? As this is the start of a new paragraph I am not sure to what this is referring.             

Response Point 12: We eliminated this sentence which is confusing. L420

Point 13: line 377 “Therefore, it is only possible to mention that the most prevalent sequence types are ST 388 and ST8 [data not shown],” In the preceeding line the authors indicate whole genome sequencing was done but was for some reason unavailable due to COVID -19 , then go on to say the most prevalent sequence types are ST388 and ST8?? Is this to suggest these are the most prevalent types from other studies, because they are not the most commonly recovered genotypes of L. monocytogenes, is this referring to a specific study of a specific source type of L. monocytogenes? If so this whole section needs clarification, better citation as well as the inclusion of more studies of this type as there are many now if they wish to incorporate these types of data.

Response Point 13: We have only limited WGS information about our strains; the lab in Austria has only sent us a file with the result of Malditof Biotyper, WGS-Nummer, MRL-Nummer, WGS, and ST.  The information about genome sequencing is not available yet.  As for the STs, we mention that ST 388 and ST 8 have been found in salmon and poultry producers, RTE food often used in sandwiches, salads, and food preparation. Ciolacu et al. [2015] also detected ST 8 in marinated fish.  Knudsen et al. [2017] suggest that L. monocytogenes ST 8 is a type of sequence that persists in food processing plants; this aspect is also corroborated by Wang et al. [2015] in RTE food processing plants in China. We added this information in the manuscript. L433-438

 Point 14: line 385 “Although PFGE is the gold standard of molecular typing, it has been losing renown due to WGS.” In this day and age the gold standard of molecular typing is most obviously whole genome sequencing. While it was PFGE at one point, that is definitely no longer the case as PFGE has several notable limitations. This should be reworded or omitted as it is incorrect.    

Response Point 14: We agree with the reviewer. Mamber et al. [2020] informed that, in coordination with the Centers for Disease Control and Prevention (CDC) PulseNet, FSIS suspended PFGE for Lm and, as of January 15, 2018, started generating Lm characterization through whole genome sequencing (WGS). In this recently published article, members of the CDC use PFGE for genotyping and suggest that, as in the 2018 study, data generated by WGS will be used in the future.

Point 15: line 400 “the duplicate strain sent for WGS did not show a mixture of strains” It was earlier indicated that WGS data were not available, then how is it known whether or not the WGS data showed a mixture of strains???              

Response Point 15: The lab in Austria sent us only one file specifying the result of Malditof Biotyper, WGS-Nummer, MRL-Nummer, WGS, and ST. In addition, they reported that no mixture of strains was found. To date, despite many emails, they have not sent us more information.

Reviewer 2 Report

The manuscript is much improved.  Attention is suggested on the following points:

  1. Point 1. If only 8 isolates were chosen for characterization, this needs the made clear in Abstract—e.g., “A small subset (n=8) of the isolates were further characterized for….”
  2. Point 2. In Abstract, it will be useful if a brief explanation is provided for “minimally” and “moderately”, and if the numbers in each category are included.  Even though this is now explained in Introduction, it would be useful for the information to be concisely presented in the Abstract also.
  3. L234-5, and Table 3. Indicate at which predicted amino acid residues the PMSCs are noted in inlA of the 8 strains.  It will be useful to employ the same terminology as in other studies, to allow cross-comparisons.  Also, it will be valuable if the specific PMSCs are indicated in a column to be added in to Table 3.  Pls note, PMSCs in inlA are uncommon in serotype 4b, and the  serotype / PMSC for the strain that is listed as 4b should be carefully checked for accuracy.  It is possible that the serotype was 1/2b, not 1/2a (see comment 6 below).  If the description is accurate, i.e., PMSC in a serotype 4b strain, it should be discussed.
  4. In Abstract, authors indicate that strains were “identified by RT-PCR”. However, reverse transcription PCR (RT-PCR) was not used, instead there is one mention of qRT-PCR (where RT is “real-time”).  Pls, amend the Abstract.  Also, it  would not be clear to the readers why qRT-PCR was used for identification, unless quantitative measures were desired.  Authors may wish to briefly explain in section 2.5.
  5. 146-7. In what way were strains  Escherichia coli ATCC 25922 and monocytogenes ATCC 7644 used as “reference strains”?   Please, clarify.
  6. 159-160. Authors write: “To identify the presumptive serotype, amplified products were sent for sequencing to Macrogen, Korea.”.  It is not clear how this would identify putative serotype.  Typically, such sequencing would differentiate between lineages, but not between serotypes within each lineage.  Thus, it would be unlikely to accurately differentiate between serotype 1/2b and 4b, or between 1/2a and 1/2c.   Multiplex PCR as described in many publications should be employed for better accuracy of serotype designations.

Author Response

Point 1: Point 1. If only 8 isolates were chosen for characterization, this needs the made clear in Abstract—e.g., “A small subset (n=8) of the isolates were further characterized for….”

Response Point 1:  The following was added: A small subset (n = 8) of the strains were further characterized for evaluation. L29

Point 2: In Abstract, it will be useful if a brief explanation is provided for “minimally” and “moderately”, and if the numbers in each category are included.  Even though this is now explained in Introduction, it would be useful for the information to be concisely presented in the Abstract also.

Response Point 2: A brief explanation was included. L26-27

Point 3: L234-5, and Table 3. Indicate at which predicted amino acid residues the PMSCs are noted in inlA of the 8 strains.  It will be useful to employ the same terminology as in other studies, to allow cross-comparisons.  Also, it will be valuable if the specific PMSCs are indicated in a column to be added in to Table 3.  Pls note, PMSCs in inlA are uncommon in serotype 4b, and the  serotype / PMSC for the strain that is listed as 4b should be carefully checked for accuracy.  It is possible that the serotype was 1/2b, not 1/2a (see comment 6 below).  If the description is accurate, i.e., PMSC in a serotype 4b strain, it should be discussed.

Response Point 3: We thank the reviewer for mentioning this information, which we used to delve in the bioinformatics of the inlA gene sequences of the strains under study. This was undertaken by two members of our international research group. Using the same terminology found in other studies, we identified that all the strains exhibited different mutations associated with changes in amino acids. However, we were only able to identify one type of mutation known as Type 6 in strain 1259. In addition, we found 14 mutations that we were unable to identify because they have not been described or published yet; we therefore propose that they are new mutations. To illustrate this, we added Table 4 and Figure 3 for better reporting.

L230-237

Table 4: L259

Figure 3: L265

Regarding the strain with the 4b serotype, which is not frequently in mutations, we added the following paragraph.  Only in publications since 2019 do PMSC appear in the inlA gene in serotype 4b. Zamani et al. [2019] found a strain of L. monocytogenes serotype 4b with PMSC. This strain had PMSC in nucleotide 1380, truncated inlA gene, and replacement mutation G → A; it is categorized as PMSC number 8. Upham et al. [2019] genomically analyzed clinical and food strains and detected that the prevalence of the inlA gene PMSC mutations in genomes of food strains was significantly higher (P < 0.0001) than in clinical strains. However, a three-codon deletion in the inlA gene associated with a highly invasive phenotype was more prevalent in genomes from clinical strains, mainly from serotype 4b, than from food strains (P < 0.001). These authors also suggest that the stress survival islet 1(SSI 1) and the inlA gene play a very important role in the evolution of L. monocytogenes strains, thus generating a more virulent phenotype that is represented by serotype 4b clinical strains or strains from a persistent environment consisting of serotype 1/2a. These aspects encountered in the literature are consistent with our reported findings; further in-depth study is required.  L393-414

Point 4: In Abstract, authors indicate that strains were “identified by RT-PCR”. However, reverse transcription PCR (RT-PCR) was not used, instead there is one mention of qRT-PCR (where RT is “real-time”).  Pls, amend the Abstract.  Also, it  would not be clear to the readers why qRT-PCR was used for identification, unless quantitative measures were desired.  Authors may wish to briefly explain in section 2.5.

Response Point 4: We used real-time PCR (RT-PCR) in our study. Current literature has used the abbreviation qPCR as an alternative to real time-PCR to avoid confusion with the reverse transcription PCR method (RT-PCR).  L29; L124-125

Real-time PCR is widely used for qualitative results (presence or absence) as in the case of our study. It is also used for quantitative purposes, pathogen genotyping, and gene expression among many other more recent areas of research [Navarro et al., 2015] https://doi.org/10.1016/j.cca.2014.10.017

Point 5: 146-7. In what way were strains  Escherichia coli ATCC 25922 and monocytogenes ATCC 7644 used as “reference strains”?   Please, clarify.        

Response Point 5: The antibiotic susceptibility test established by CLSI 2018 specifies cut-off points for antibiotics.  The Escherichia coli ATCC 25922 and L. monocytogenes ATCC 7644 strains were used as references in our study. This has been widely used and reported. For example, Pasavento et al. [2010], Santos Oliveira et al. [2018], and Caruso et al. [2020] have used ATTC stains from E. coli, Listeria, Streptococcus, and Pseudomonas to compare their cut-off points, as in our study.

Point 6: 159-160. Authors write: “To identify the presumptive serotype, amplified products were sent for sequencing to Macrogen, Korea.”.  It is not clear how this would identify putative serotype.  Typically, such sequencing would differentiate between lineages, but not between serotypes within each lineage.  Thus, it would be unlikely to accurately differentiate between serotype 1/2b and 4b, or between 1/2a and 1/2c.   Multiplex PCR as described in many publications should be employed for better accuracy of serotype designations.

Response Point 6: We used the concept of presumptive serotype when waiting for the WGS results. The DNA and all the amplifications were sent; we received the bioinformatic analysis from Macrogen, which confirmed the strains as L. monocytogenes and reported their serotype.